# Using RE-AIM to Assess Infant Early Childhood Mental Health Practices in Classrooms Serving Children with and Without Disabilities

**DOI:** 10.3390/healthcare12242501

**Published:** 2024-12-11

**Authors:** Ruby Natale, Tara Kenworthy LaMarca, Tanha Rahman, Elizabeth Howe, Rebecca J. Bulotsky-Shearer, Yaray Agosto, Jason Jent

**Affiliations:** 1Mailman Center for Child Development, University of Miami Miller School of Medicine, Miami, FL 33136, USA; tlk38@miami.edu (T.K.L.); txr321@med.miami.edu (T.R.); exh706@med.miami.edu (E.H.); yagosto@med.miami.edu (Y.A.); jjent@med.miami.edu (J.J.); 2Department of Psychology, University of Miami, Coral Gables, FL 33146, USA; rshearer@miami.edu

**Keywords:** infant and early childhood mental health consultation, disabilities, RE-AIM, teachers, classroom

## Abstract

**Background/Objectives:** High-quality inclusive education is important for promoting the positive development of children with disabilities in early childhood care and education (ECCE) settings. However, ECCE teachers may not have the knowledge and skills to manage challenging behaviors in young children, especially those with disabilities. Infant and Early childhood mental health consultation (IECMHC) is one mechanism to support the professional development of teachers. This study explored the impact of an evidence-based IECMHC program, Jump Start Plus COVID Support (JS+CS), on outcomes for teachers in classrooms including children with disabilities. **Methods:** Utilizing a RE-AIM framework, we examined the extent that JS+CS impacted teacher outcomes related to classroom practice and teacher attitudes after the initial intervention period. In addition, we examined the extent that the classroom children with disability ratio moderated the impact of the intervention on teacher outcomes. Using a cluster randomized controlled trial in a sample of 138 racially and ethnically diverse teachers in 31 ECCE centers, we examined the reach, effectiveness, adoption, and implementation of JS+CS. **Results:** The results indicate that the JS+CS program shows promise as an intervention to support ECCE teachers working in classrooms with children with disabilities, particularly in improving teacher safety practices, behavior management skills, and resiliency coping. In addition, the program was adopted equally in classrooms that served children with and without disabilities. **Conclusions:** This is a unique contribution to the literature given that no previous IECMHC programs have examined adoption in classrooms serving children with disabilities. Further investigation is needed to determine the specific factors that impact program implementation considering that this study was conducted during various phases of the COVID-19 pandemic.

## 1. Introduction

### 1.1. Behavior Problems in Early Childhood Care and Education (ECCE) Centers

Young children aged from birth to five spend most of their time within ECCE programs, including those with disabilities. Inclusive ECCE programs are critical settings for educating all young children, including those with disabilities [1,2]. ECCE programs are also vital social determinants of health, as they allow for parents to work while knowing their child is being cared for [3]. ECCE programs implement evidence-based curricula that support young children’s development, offset risk factors such as behavior problems, and prepare children for kindergarten.

Teachers in ECCE programs often face challenges with child behaviors. There is a high incidence of young children displaying behaviors that are considered challenging and problematic for adults [4,5]. High rates can be attributed to the normal progression of a child’s development; however, incidences of challenging behaviors can also be co-occurring with an identified disability or a sign of a suspected disability [6]. Teachers in ECCE programs may not have the knowledge and skills for decreasing incidences of these behaviors in young children, particularly those with identified or suspected disabilities [6]. Additionally, they may not be presented with opportunities for comprehensive professional development that can support their adoption of evidence-based practices for promoting young children’s social competence and preventing challenging behaviors [7,8]. Addressing young children’s challenging behavior is frequently cited as contributing to higher levels of teacher stress and is a primary need for teacher professional development [9,10]. Infant and early childhood mental health consultation (IECMHC) is an approach with a growing body of evidence of its effectiveness for supporting teachers in adopting classroom and practices individualized to the child for promoting children’s social competence and decreasing incidences of challenging behaviors [11,12,13].

### 1.2. Infant and Early Childhood Mental Health Consultation

IECMHC strengthens the abilities, knowledge, and access to resources of ECCE professionals, so they can create healthier environments and better support the children in their care [14,15,16]. In doing so, IECMHC aims to support early action to prevent potential mental health or developmental issues in young children before they arise. IECMHC programs are unique in that the focus of the intervention is not on the child but rather the program and teacher, supporting them to utilize evidence-based, child-level interventions to prevent challenging behaviors and promote social competence. IECMHC programs can support children with identified or suspected disabilities by supporting teachers to implement both classroom-wide and individualized practices that encourage children’s full participation in classroom routines. Classroom-wide practices include the use of visual schedules and transition routines. Individualized supports include the use of assistive technology and the use of individualized behavioral intervention plans.

Studies on IECMHC have demonstrated its effectiveness for improving teacher- and classroom-level outcomes. A recent systematic review of studies of the preceding decade found that seven studies described positive teacher-level outcomes following IECMHC, including job stress and teacher efficacy [17]. The classroom-level outcomes of this review were mixed, with two of three studies reviewed finding improvements in classroom quality in IECMHC classrooms. Some more recent studies support the positive impact of IECMHC at the classroom level, including for teachers’ classroom management and the classroom environment or climate [18,19]. Furthermore, there is strong evidence that classrooms where teachers receive IECMHC have reduced teacher stress [12] and improved child developmental and behavioral outcomes compared to those that do not [18]. These findings underscore the effectiveness of IECMHC in improving ECCE classroom environments and positive developmental outcomes, giving credence to the idea that these models could be effective when implemented in classrooms with children with disabilities.

One of the few studies examining the implementation and outcomes of IECMHC among children with disabilities was conducted in Head Start programs, which are federally funded programs that support school readiness for the children of low-income families from birth to age five through services centered on cognitive, social, and emotional development. There is evidence that children with multiple disabilities in Head Start programs perform better with regard to academic and language skills by kindergarten compared to those with multiple disabilities who do not participate in Head Start [20]. Assessing the impact of IECMHC models on children with and without disabilities ensures that consultation models are promoting equitable outcomes for all children [11].

### 1.3. Jump Start Plus COVID Support

Jump Start is an evidence-based IECMHC program, based on the Georgetown Model, that is effective for increasing teachers’ use of strategies that promote positive child behavior and reducing a child’s risk of expulsion [21]. At the outset of the COVID-19 pandemic, our research team implemented an exploratory study to modify the Jump Start intervention, resulting in the development of Jump Start Plus COVID Support (JS+CS), which promoted social–emotional development in the context of COVID-19 [3]. JS+CS centers on four pillars of support: safety, communication, self-care, and trauma-informed behavior support. Given COVID-19-related school closures, both in-person and virtual consultations were available [22]. However, it remained unclear how this intervention would impact the classroom practices and attitudes of teachers with various ratios of children with and without disabilities.

### 1.4. Measuring Implementation

To understand how JS+CS impacted teachers in diverse ECCE settings serving children with and without disabilities, we applied an implementation science framework. Implementation science is “the application and integration of research evidence into practice” [23]. In other words, implementation science deepens the understanding of what it takes to apply evidence-based practices in real-world settings at scale. JS+CS has been implemented in many types of programs stemming from different state and federal funding sources (e.g., Head Start, Child Care and Development Fund) and has reached diverse children, families, and teachers (e.g., race, ethnicity, ability). Therefore, it is important to understand how and why JS+CS works in diverse settings, for diverse populations.

The Reach, Effectiveness, Adoption, Implementation, Maintenance (RE-AIM) [24] framework is a widely used implementation science framework that orients researchers to elements essential for planning and evaluating the practitioner use of evidence-based interventions, to understand how they work in real-world settings [25,26]. RE-AIM has been applied to evaluate the outcomes of a prior iteration of the Jump Start program, which focused on consultation with teachers and direct short-term intervention with children [21]. Therefore, given the extensive use of RE-AIM in community settings and its use specifically with the Jump Start program, this framework was determined to be appropriate for evaluating the JS+CS intervention (a Type 1 hybrid effectiveness–implementation approach) as well, with a specific focus on reach, adoption, effectiveness, and implementation through the following research questions:Reach: Who did this intervention reach?Effectiveness: Do centers that received JS+CS have improved teacher outcomes compared to active controls?Adoption: Does the child disability ratio in classrooms moderate the effectiveness of the intervention on teacher outcomes for centers receiving JS+CS compared to active controls (see Figure 1)?Implementation: Did the intervention and active control receive the expected intervention dosage?

## 2. Materials and Methods

### 2.1. Setting

This study took place in ECCE centers across south Florida serving historically minoritized communities from September 2021 to September 2023. As described previously [3], all centers were part of a Quality Improvement System, a governing body that enrolls childcare centers and monitors the quality of their services. The center inclusion criteria were (1) having ≥50 children (≥30 of whom were 18 months–3 years old); (2) being located in the low-income census tract, with at least 50% of families receiving a childcare subsidy; (3) serving at least 60% Hispanic or 60% Non-Hispanic Black families; (4) having directors, teachers, and parents who agreed to participate; and (5) not having previously enrolled in an early childhood mental health consultation program.

Children within the sample included those who participated in either Part C or Part B Section 619 of the Individuals with Disabilities Education Act (IDEA) [27]. IDEA regulates special education services to children with disabilities. IDEA Part C is for children under the age of three and includes formalized learning plans, i.e., individualized family service plans (IFSPs), that describe the child’s and family’s early intervention needs. Children ages three to five participate in the IDEA Part B, which includes individualized education programs (IEPs) that dictate their special education goals and objectives. Children with suspected disabilities may be involved in a referral process that could include developmental screenings, referral for an evaluation under the IDEA, or pending the results of an evaluation and a formal recommendation for IDEA services. In the state of Florida, approximately 2.4% of children receive early intervention through Part C of the IDEA, and an additional 5.4% of preschool age children receive special education services through Part B of the IDEA [2]. The majority of children under Part B receive their special education services outside of the ECCE classroom [2], meaning that ECCE teachers cannot easily access training from certified therapists with the knowledge and skills to deliver effective interventions. Therefore, implementing an IECMHC model in general ECCE settings, with teachers benefitting from direct consultation to examine changes in children’s social emotional skills, is critical. In this sample, teachers’ classrooms with children with an identified or suspected disability ranged from 0% to 75% (*M* = 8.86%, *SD* = 15.04%).

### 2.2. Participants

The sample comprised 138 teachers in 31 early childcare centers. Randomization of teachers to interventions at the childcare center level largely succeeded, except for the random distribution of teacher ethnicity across centers. Further description of the participants is provided within Section 3.1.

### 2.3. Measures

Each teacher completed a sociodemographic questionnaire that asked questions regarding general demographic information about their social identity.

The Health Environment Rating Scale-Classroom (HERS-C) is a 30 min classroom observation developed by the study investigators. It comprises four domains: safety, behavioral supports, communication, and resiliency coping. These domains align with core national standards for health and safety in ECCE programs [28], map on to the JS+CS pillars, and allow for the measurement of expected areas of change in the control centers’ obesity prevention intervention (nutrition and physical activity). This measure is scored on a 7-point Likert scale rated from “little or no implementation” (1) to “excellent implementation” (7). In this study, the internal consistency for the HERS-C was good (α = 0.82). All four pillars were examined.

The Childcare Worker Job Stress Inventory is a measure of workplace stress for childcare center workers. Within the current study, the Job Control scale was utilized, in which higher scores suggest that teachers have increased autonomy with decision making related to their classrooms, collaboration with parents, and time off. The Job Control scale contains 17 items rated on a Likert scale, rated from “very little” (1) to “very much” (5). Adequate validity and reliability of this scale have been established [29]. In this study, the internal consistency for the Childcare Worker Job Stress Inventory Job Control Scale was good (α = 0.85).

The Teacher Opinions Survey is a self-report measure of teachers’ confidence in their ability to manage challenging child behaviors [30]. This 12-item measure is rated on a Likert scale from “strongly disagree” (1) to “strongly agree” (5). Higher scores on this measure represent higher perceptions of teachers’ confidence in managing children’s behaviors. The total score for this measure had adequate internal consistency (α = 0.79).

Child disability classroom ratio was calculated as the number of children with identified or suspected disabilities (as reported by teachers) divided by the total number of children within the classroom.

Consultation dosage was calculated as the number of total consultation sessions each teacher was recorded as receiving during the intervention period.

### 2.4. Procedures

This study was approved by the University of Miami’s Institutional Review Board and is currently registered with http://ClinicalTrials.gov number NCT05445518. Details of the study design and procedures were previously described [3]. A cluster-randomized trial was used to evaluate the effectiveness of the JS+CS program and understand the barriers and facilitators involved in its implementation. As such, centers were assigned to different groups, and outcomes were measured for individuals within those centers. Thirty-one childcare centers were randomly assigned to either the intervention group or an attention control group; the JS+CS program was introduced to the intervention group (16 centers) while the attention control group (15 centers) participated in an alternate Healthy Caregivers-Healthy Children (HC2) obesity prevention program.

Recruitment focused on 31 childcare centers that served low-income families from ethnic minority groups. Interested and eligible centers provided informed consent alongside completing a baseline survey. Graduate-level research assistants from similar ethnic backgrounds as the participants managed the recruitment and consent process. Following this phase, virtual kick-off sessions introduced the programs and telepresence robot to teachers and parents. Data were collected at baseline and after the intervention (14 weeks) using electronic surveys.

#### 2.4.1. Intervention Group

The intervention was delivered via three steps as follows:

Step 1: Toolkit Implementation. The program focused on four pillars—self-care, behavior support, safety, and communication—over six weeks, and all teachers in the intervention group received a comprehensive toolkit that included an action plan outlining weekly goals and strategies to accomplish those objectives. The toolkit also featured 24 infographics highlighting facts and addressing misconceptions, as well as cartoon videos demonstrating coping strategies for educators and directors to combat stressors. The toolkit further contained some children’s content, including infectious disease prevention videos that taught best practices like hand washing. Teachers were able to receive real-time feedback during this process from mental health consultants (MHCs) who provided 14 weeks of teleconsultations (MHCs dedicated 4 h weekly per center, split between directors and teachers). Consultants could provide consultation on up to three infographics per consultation, indicating that the minimum number of consultations provided to teachers was 8 sessions. Throughout this first phase, directors received an hour of consultation weekly to build capacity center-wide, while teachers received an hour of group consultation +20 min of individual consultation per week to integrate the four pillars into their daily routines. Each school was equipped with a Kubi Plus telepresence (Xandex Inc., Petaluma, CA, USA) robot to facilitate this through enabling MHCs to deliver the intervention remotely and offer live feedback.

Step 2: Sustainability and Cultural Relevance. To ensure the sustainability of the program, a lead teacher known as the “Program Champion” was assigned at each center and given an annual USD 500 stipend for their role in helping to support the ongoing implementation of toolkit strategies. Mental health consultants (MHCs) provided quarterly 2 h refresher training sessions in the second and third years of this study to reinforce and enhance teachers’ skills. The toolkit was adapted using the Ecological Validity Model [31] to ensure cultural relevance with translations into Spanish and the inclusion of photos and data that would be relevant to ethnic minorities.

Step 3: Monitoring Fidelity. The JS+CS intervention involved tracking the quality of intervention delivery, participant responsiveness, adherence, exposure, and program differentiation to ensure the intervention was implemented as intended.

#### 2.4.2. Control Group

The control group received the Healthy Caregivers-Healthy Children (HC2) obesity prevention program, following the same 3-step process and receiving the same pre- and post-measures and incentives as the intervention group. The HC2 program, developed and implemented through federal funding, has been found to improve obesity prevention practices within centers catering to similar ethnic populations. The HC2 program was also approved by the Institutional Review Board (IRB) approval and registered with ClinicalTrials.gov (NCT05445518).

### 2.5. Analysis

Reach. Descriptive statistics were calculated for all sociodemographic variables within SPSS 29.0 (Armonk, NY, USA). Chi-square analyses were conducted to determine if there were any significant differences between intervention groups in terms of teacher sociodemographic characteristics. A significant difference was found in teacher ethnicity but, because of the small cell size, this variable was not included in the subsequent analyses.

Effectiveness and Adoption. To examine the predictors of teacher outcomes, we conducted multilevel modeling analyses using Mplus version 8.10. Missing data were handled using full information maximum likelihood (FIML). This approach allowed for the inclusion of all available data, potentially reducing bias and increasing statistical power compared to listwise deletion [32]. The data were analyzed following recommended procedures [33,34]. We examined whether intervention group assignment predicted teacher outcomes (effectiveness) while examining whether the child disability ratio within teacher classrooms moderated the relationship between intervention group assignment and teacher outcomes (adoption). In alignment with best practices [33], these analyses proceeded in a series of several steps. First, data were nested at the childcare center, as the intervention was randomly assigned at the program level. Therefore, for each outcome variable, we calculated intraclass correlation coefficients (ICCs). An empty (null) two-level model was specified with childcare center as the clustering variable. The teacher outcome variable was included at the classroom level. A statistically significant ICC indicated that significant variance in an outcome was attributable to differences between childcare centers and that such nesting effects should be accounted for in subsequent modeling. The ICCs for various measures indicated significant clustering effects at the program level for most variables, with ICCs ranging from 0.084 to 0.842 and *p*-values largely below 0.05. Specifically, HERS-C Communication (ICC = 0.650, *p* = 0.001), HERS-C Behavior (ICC = 0.645, *p* = 0.007), HERS-C Safety (ICC = 0.303, *p* = 0.022), HERS-C Resilience (ICC = 0.842, *p* = 0.001), Teacher Resiliency (ICC = 0.185, *p* = 0.041), Teacher Efficacy (ICC = 0.343, *p* = 0.015), Childcare Worker Job Stress Inventory Job Demand Scale (ICC = 0.32, *p* < 0.001), and Childcare Worker Job Stress Inventory Job Resources Scale (ICC = 0.084, *p* < 0.001) all showed significant clustering, while the Childcare Worker Job Stress Inventory Job Control Scale (ICC = 0.191, *p* = 0.089) did not reach statistical significance but still showed a notable clustering effect; thus, multilevel modeling controlling for nesting at the childcare center level was justified for these analyses. Therefore, in the subsequent models, we accounted for nesting at the childcare center level and the classroom level (2-level model).

Second, after investigating the ICCs, we used multilevel modeling in Mplus to examine whether the intervention group (center level) predicted teacher outcomes (i.e., HERS-C outcomes, Teacher Resilience, Teacher Self-Efficacy, and Childcare Worker Stress). We also examined the extent that the child disability ratio moderated the relationship between intervention group and teacher outcomes (classroom level). The interaction between child disability ratio and intervention was examined at the classroom level to capture classroom-specific variations in intervention effectiveness, preserve important classroom-level information, and maintain consistency with other classroom-level predictors. This approach allowed for a more nuanced understanding of how individual teachers’ experiences with the intervention might vary based on their specific classroom composition while also providing greater statistical power to detect potentially subtle interaction effects [35]. By analyzing this interaction at the classroom level rather than aggregating to the program level, the model could more accurately represent the immediate contextual factors that influenced the intervention’s impact on teacher outcomes. To account for the nested structure of teachers within programs within each analysis, we employed a two-level random effects model using maximum likelihood estimation with Monte Carlo integration (500 iterations). At Level 1 (within-program level), we included teachers’ baseline classroom variables (group-mean-centered), child disability ratio, and the interaction between intervention condition and child disability ratio (group-mean-centered). At Level 2 (between-program level), we included intervention condition (grand-mean-centered) as a program-level predictor. These estimation procedures ensured all responses, even those with partially missing data, were included in study analyses [33]. The use of such estimation procedures ensured that an intention-to-treat framework was used in all analyses that included all teachers who began the study, even if they did not complete the intervention. Using an intention-to-treat approach ensures outcome estimates are not biased by only including participants who completed the intervention [36]. To control for clustering of teachers within childcare centers, the “Cluster=” command was used, and childcare center was included as the clustering variable. 

Implementation. Descriptive statistics and one-way ANOVAs were calculated with SPSS 29.0 for intervention group consultation dosage.

## 3. Results

### 3.1. Reach

A total of 31 early childcare center programs and 138 teachers participated in this study. Teachers were predominantly Hispanic and Spanish-speaking. See Table 1 for additional demographic information. The demographics reflected the local community in Miami-Dade County, FL, where, per the 2020 Census, 68.7% of the population was Hispanic, 14% was Black, and 66.3% spoke Spanish at home [37]. Compared to the center-based ECCE workforce nationally, our sample over-represented Hispanic and under-represented Black workers. A national survey in 2020 indicated that 14% of center-based ECCE teachers and caregivers were Hispanic, 17% were Black, and the majority (63%) were non-Hispanic White; 9% of teachers and caregivers in centers spoke a language other than English [38].

### 3.2. Effectiveness and Adoption

#### 3.2.1. HERS-C Safety

Multilevel modeling was conducted to determine the predictors of teacher outcomes and to determine if the child disability ratio moderated the relationship between intervention group and teacher outcomes. With regard to HERS-C-Safety, a multilevel model was used to examine the effects of the intervention, child disability ratio within a classroom, and their interaction on teachers’ use of safety practices within the classroom, while controlling for baseline levels (see Table 2). The model accounted for the nested structure of teachers within the programs. At the classroom level, baseline teacher safety practices significantly predicted the post-intervention scores. However, neither child disability ratio nor the interaction between child disability ratio and intervention were significant predictors of teacher safety practices after the intervention. At the childcare center level, being assigned to JS+CS significantly predicted increased use of teacher safety practices after the intervention. The model explained 13.5% (*R*^2^ = 0.135, *p* = 0.111) of the classroom-level variance and 49.0% (*R*^2^ = 0.490, *p* = 0.026) of the childcare center -level variance in support for children’s social–emotional competence at follow-up.

#### 3.2.2. HERS-C Behavior

Our study employed a multilevel modeling approach to examine the effects of JS+CS on observed teacher behavior management skills while accounting for baseline levels, child disability ratio within the classroom, and the interaction between the intervention and disability ratio. The results of our analysis revealed significant effects at both the classroom and childcare center levels (see Table 3). At the classroom level, baseline-observed teacher behavior management practices significantly predicted post-intervention teacher practices. However, neither the child disability ratio nor its interaction with the intervention group significantly predicted the observed teacher behavior management practices. Our model explained 17.2% of the classroom-level variance (*R*^2^ = 0.172, SE = 0.089, *p* = 0.053). The nonsignificant effects of the child disability ratio and its interaction with treatment suggested that the proportion of children with disabilities in a classroom did not substantially influence teacher behavior management practices nor did it moderate the effect of the intervention. At the childcare center level, we found a significant effect of JS+CS on the observed teacher behavior management practices. The treatment explained approximately 32.0% of the variance between intervention groups (*R*^2^ = 0.320, SE = 0.173, *p* = 0.065).

#### 3.2.3. HERS-C Communication

The effects of JS+CS on the observed teacher communication practices after the intervention, while considering the influence of baseline community practices, child disability ratio, and the interaction between intervention group and disability ratio, were explored through multilevel modeling. The results from our multilevel analysis revealed significant effects at the classroom level but not at the between level (see Table 4). At the classroom level, the classroom child disability ratio significantly predicted the observed post-intervention teacher communication practices. That is, across intervention groups, teachers with higher numbers of children with disabilities within their classroom were associated with less observed bidirectional communication with caregivers and fewer interactions that reflected cultural competence, including awareness of implicit biases, incorporation of diverse representation in classroom materials, and adaptability to students’ varied cultural, linguistic, and disability needs. At the childcare center level, we found no significant effect of the intervention on teacher communication practices. Our model explained 17.6% of the classroom-level variance (*R*^2^ = 0.176, *p* = 0.068) in teacher communication practices. However, at the childcare center level, the model explained only 8.2% of the variance (*R*^2^ = 0.082, *p* = 0.473), which was not statistically significant.

#### 3.2.4. HERS-C Resiliency

This multilevel analysis examined the effects of the intervention on observed post-intervention teachers’ resiliency coping, while accounting for the baseline observed resiliency coping, classroom child disability ratio, and the interaction between intervention and disability ratio (see Table 5). At the classroom level, baseline observed teacher resiliency significantly predicted the observed post-intervention teacher resiliency coping. However, the classroom child disability ratio and its interaction with intervention did not significantly predict the observed post-intervention teacher resiliency coping. At the childcare center level, we found a significant effect of JS+CS on observed post-intervention teachers’ resiliency coping. The models were not statistically significant, with 15.2% of the within-level variance (*R*^2^ = 0.152, *p* = 0.108) and 19.8% of the between-level variance (*R*^2^ = 0.198, *p* = 0.180) in the observed post-intervention teachers’ resiliency coping, suggesting caution in interpreting the overall model fit.

#### 3.2.5. Teacher Opinion Survey

A multilevel model was conducted to examine the effects of classroom child disability ratio, intervention, and their interaction on teachers’ self-efficacy, while controlling for baseline levels (see Table 6). At the classroom level, neither baseline teacher efficacy, child disability ratio, nor the interaction between classroom child disability ratio and intervention significantly predicted teacher efficacy after the intervention. At the childcare center level, the intervention did not significantly predict differences in teacher efficacy between programs.

#### 3.2.6. Childcare Worker Job Stress Inventory: Job Control

A multilevel model was applied to examine the effects of classroom child disability ratio, intervention, and their interaction on childcare workers’ job control, while controlling for baseline levels (see Table 7). At the classroom level, baseline teacher job control significantly predicted post-intervention job control, indicating that childcare workers who reported higher levels of job control at baseline tended to maintain higher levels post-intervention. However, neither classroom child disability ratio within the classroom nor its interaction with the intervention significantly predicted job control post-intervention. At the childcare center level, the intervention did not significantly predict differences in job control between intervention groups. The model explained 6.5% of the classroom-level variance and 7.9% of the childcare center-level variance in childcare worker job control at follow-up, though neither of these *R*^2^ values were statistically significant (*p* = 0.152 and *p* = 0.603, respectively).

### 3.3. Implementation

An analysis of variance (ANOVA) was conducted to compare the number of consultation sessions received by teachers in the JS+CS group and the HC2 group. The results revealed a statistically significant difference between the two groups (*F* (1, 134) = 4.664, *p* = 0.033). Specifically, teachers in the JS+CS group (*M* = 11.75, *SD* = 9.85) received a higher number of consultation sessions compared to teachers in the HC2 group (*M* = 8.82, *SD* = 5.64). Of note, within both intervention groups, teachers were supposed to receive a minimum of eight consultations over a 14-week period. However, the number of teachers that reached this threshold varied by group. Within JS+CS, 31.7% of participants dropped immediately (zero consults), 3.2% of participants received between one and seven consultations (below the required minimum dosage), and 65.1% of participants received the minimum dosage of eight or more consultations. Within HC2, 17.8% of participants dropped immediately (zero consults), 24.7% of participants received between one and seven consultations (below the required minimum dosage), and 57.5% of participants received eight or more consultations.

## 4. Discussion

This study utilized the RE-AIM framework to evaluate the implementation and effectiveness of the Jump Start Plus COVID Support (JS+CS) program, an IECMHC intervention in ECCE centers serving diverse populations including children with and without disabilities. Our findings provide valuable insights into the reach, effectiveness, adoption, and implementation of JS+CS in real-world settings.

### 4.1. Reach

The JS+CS program successfully reached a diverse sample of 138 teachers across 31 ECCE centers, predominantly serving Hispanic and Spanish-speaking communities. This demographic aligns with the target population and reflects the community composition, suggesting good reach within the intended audience. However, the sample included fewer Black teachers compared to the national childcare worker population [39], and the geographic region where this study was conducted may limit the generalizability of the findings to more diverse settings. The high proportion of Hispanic participants (87.5%) provided a unique opportunity to understand the program’s impact on this underserved population, addressing a gap in the literature on culturally responsive early childhood interventions [11].

### 4.2. Effectiveness and Adoption

The JS+CS intervention demonstrated mixed effectiveness across various teacher outcomes. Significant positive effects for teachers within the JS+CS group were observed in teacher safety practices, behavior management skills, and resiliency coping. Teachers in the JS+CS group showed significantly improved safety practices and better observed child behavior management skills post-intervention, indicating the program’s success in enhancing classroom safety protocols and improving teachers’ ability to manage challenging behaviors. These findings align with previous research on the effectiveness of IECMHC programs in improving classroom quality and teacher-child interactions [15]. Additionally, JS+CS teachers exhibited higher levels of observed resiliency coping compared to teachers in the HC2 control group, suggesting that the intervention may have enhanced teachers’ ability to cope with workplace stressors, a crucial factor in maintaining a positive classroom environment during COVID-19 [40].

The adoption of JS+CS practices varied across outcomes and was influenced by several factors. Notably, higher classroom child disability ratios were associated with fewer observed bidirectional communications with caregivers and less culturally competent interactions. The JS+CS intervention itself did not significantly improve these practices. This finding underscores the importance of providing targeted support for teachers working with children with disabilities [41], as emphasized in the literature on inclusive early childhood education [42,43]. It is possible that teachers need additional skills to better support the needs of children with disabilities. For example, assistive technology improves outcomes for children with disabilities, but there is a need for more professional development opportunities for teachers to promote its implementation in the community [44]. The association between higher numbers of children with disabilities in classrooms and fewer bidirectional, culturally responsive and inclusive teacher communication practices could be partially attributed to increased teacher stress and workload. Previous research has found that teachers often feel unprepared to meet the complex needs of children with disabilities in inclusive settings, particularly regarding culturally responsive practices [45].

The lack of significant improvement in teacher self-efficacy and stress related to job control perceptions during the COVID-19 pandemic could be partially related to the unique challenges teachers faced during this period. Though a previous open trial of JS+CS found positive effects on improvements in stress related to job control [21], this trial was conducted prior to COVID-19. The overwhelming stress and rapid transitions that occurred in childcare centers during COVID-19 likely undermined any positive effects of the JS+CS program on these outcomes. Many teachers noted that frequent changes in federal and state health and safety protocols contributed to feelings of reduced autonomy and efficacy [46,47]. Also of note, the original Jump Start intervention [21] was designed to be a 24-week intervention that included reflective consultation on one topic per weekly session. However, due to time constraints of the cluster-randomized trial and the childcare center academic year for data collection timepoints, the intervention had to be consolidated into 14 weeks. The shortened duration of the intervention may have limited the effectiveness of JS+CS on these teacher outcomes. These findings suggest that while the JS+CS program may have had some positive effects, the extraordinary circumstances of teaching during the COVID-19 pandemic could have overshadowed improvements in self-efficacy and job control perceptions. It is possible that additional structural and system supports (e.g., protected paid leave when exposed to COVID-19, provision of and availability of high-quality personal protective equipment) were needed beyond what early childhood consultation models could provide during the study period (2021–2023) in order to successfully support teachers’ ability to feel confident in addressing rapidly changing pandemic-specific challenges and maintaining healthy perceptions of stress related to their control over their job responsibilities.

For most outcomes, baseline levels were significant predictors of post-intervention scores, suggesting that teachers’ initial skills and perceptions played a crucial role in their responsiveness to the intervention. The interaction between the intervention group and child disability ratio did not significantly predict outcomes, indicating that the effectiveness of JS+CS was not moderated by the proportion of children with disabilities in the classroom. This suggests that JS+CS was adopted equally in classrooms that served children with and without disabilities, which was an intended outcome of the program. To the best of our knowledge, no previous IECMHC programs have examined adoption in classrooms serving children with disabilities.

### 4.3. Implementation

The results revealed significant differences in consultation session engagement between the JS+CS and control groups, with JS+CS participants receiving a higher number of consultation sessions. However, both groups struggled to meet the minimum required dosage of eight consults over a 14-week period, with only 65% of JS+CS participants and 57% of control-group participants receiving the minimum dosage or more. In a related study of this cluster-randomized controlled trial [22], implementation was identified as a difficulty partially due to the unique challenges during the COVID-19 pandemic, particularly regarding the mode of consultation delivery (e.g., virtual consultation approach vs. in-person consultation approach). As COVID-19 restrictions eased, consultants from JS+CS and HC2 revealed a growing preference among both consultants and childcare staff for in-person consultations. While virtual consultations were initially adopted as a safety measure, they introduced unforeseen obstacles such as technological difficulties, scheduling issues, and competing demands on participants’ time and attention. These conflicting demands may have ultimately negatively impacted consultation dosage and ultimately minimized the expected effects on teacher outcomes. The varying preferences for consultation modality underscore the need for flexibility in program implementation and the importance of adaptable delivery methods to meet the diverse needs of childcare staff in changing circumstances [22].

### 4.4. Limitations

Several limitations should be considered when interpreting these results. The high immediate teacher attrition rate in both the JS+CS and the HC2 groups (no consultations received) limited our understanding of the effects of the intervention. Furthermore, a systematic exploration as to why teachers were lost to follow-up was not conducted. However, anecdotally, some childcare centers completely dropped from this study due to concerns about measure burden and time commitment of participating in ongoing consultation when they were already short-staffed. The predominantly Hispanic sample, while valuable for understanding this population, limits generalizability to more diverse settings. Some outcomes relied on self-report measures, which may be subject to social desirability bias. Additionally, this study focused on immediate post-intervention outcomes, and long-term effects, including those on child outcomes, remain unknown. Lastly, it should be noted that this study was conducted during the height of the COVID-19 pandemic, and centers may have thus been faced with unique challenges that may have impacted outcomes that are not generalizable to the post-pandemic era.

### 4.5. Future Directions

Future research should address these limitations by implementing strategies to improve participant retention, expanding this study to more diverse populations and settings, incorporating objective measures of teacher practices and child outcomes, and conducting follow-up data collection to assess the long-term impact of JS+CS on teacher and child outcomes. Investigating additional factors contributing to the low dosage adherence and developing strategies to improve implementation fidelity should also be prioritized. The finding that the child disability classroom ratio did not impact the intervention effects on teacher outcomes is promising. However, more specificity is needed in future research to better understand how IECMHC models individualize consultation services for teachers given that child disabilities widely range in type, severity, and impact on academic and social–emotional learning.

## 5. Conclusions

The JS+CS program represents a significant advancement in understanding how IECMHC can be implemented effectively in diverse early childhood settings, particularly during unprecedented challenges like the COVID-19 pandemic. Our findings produced three key contributions to the field. First, JS+CS demonstrated meaningful success in enhancing critical teacher competencies, specifically in safety practices, behavior management skills, and resilience coping—essential skills that became even more crucial during the pandemic. Second, JS+CS showed equivalent effectiveness across classrooms regardless of the proportion of children with disabilities, marking an important step forward in developing inclusive early childhood interventions that serve all children effectively.

Third and finally, our implementation findings revealed both opportunities and challenges in delivering IECMHC during periods of significant disruption. While the program successfully reached a predominantly Hispanic population—a traditionally underserved group in early childhood interventions—the challenges in maintaining consistent consultation dosage highlight the need for more flexible and adaptable delivery models. The preference to shift from virtual to in-person consultation as pandemic restrictions eased suggests that hybrid models might optimize future implementation.

These findings have immediate practical implications for early childhood education policy and practice. They suggest that while IECMHC programs like JS+CS can effectively support teacher competencies even during crisis periods, additional structural supports (such as protected paid leave and adequate personal protective equipment) may be necessary to fully address teacher stress and self-efficacy. Additionally, to improve the effectiveness of IECMHC in classrooms including students with disabilities, future programs should consider offering professional development specific to the needs of this population, such as assistive technology.

This study opens important avenues for future research, particularly in understanding how the IECMHC can be tailored to address the specific needs of teachers working with children with varying types and severities of disabilities. Additionally, investigating the long-term impacts of such interventions on both teacher and child outcomes remains crucial for establishing the sustained value of these programs in promoting high-quality, inclusive early childhood education.

## Figures and Tables

**Figure 1 healthcare-12-02501-f001:**
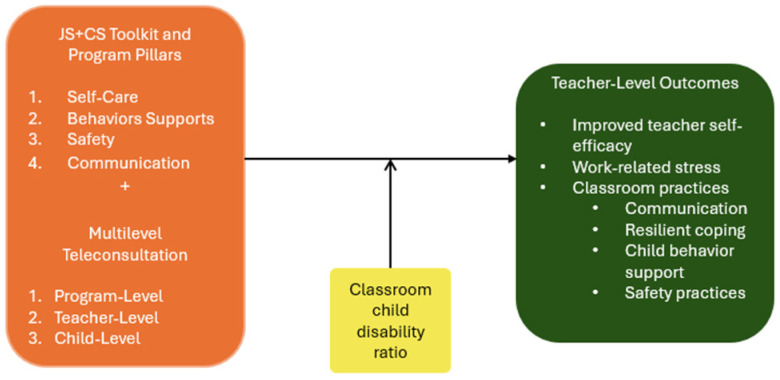
Conceptual model of JS+CS on teacher outcomes.

**Table 1 healthcare-12-02501-t001:** Participant demographics.

Variable	HC2 (n = 75)	JS+CS (n = 63)	Total (N = 138)	Test Statistic	*p*-Value
Age, Mean (SD)	44.06 (12.71)	45.00 (12.31)	44.49 (12.49)	*F* = 0.191	0.662
Sex				*χ*^2^ = 1.199	0.273
Female	75 (100.0%)	62 (98.4%)	137 (99.3%)		
Male	0 (0.0%)	1 (1.6%)	1 (0.7%)		
Race				*χ*^2^ = 1.498	0.827
White	59 (80.8%)	52 (82.5%)	111 (81.6%)		
Black	9 (12.3%)	7 (11.1%)	16 (11.8%)		
Native American	1 (1.4%)	0 (0.0%)	1 (0.7%)		
Multiracial	2 (2.7%)	3 (4.8%)	5 (3.7%)		
Other	2 (2.7%)	1 (1.6%)	3 (2.2%)		
Ethnicity				*χ*^2^ = 9.476	0.05
Hispanic	65 (86.7%)	54 (88.5%)	119 (87.5%)		
Non-Hispanic White	4 (5.3%)	1 (1.6%)	5 (3.7%)		
Non-Hispanic Black	2 (2.7%)	2 (3.3%)	4 (2.9%)		
Haitian	4 (5.3%)	0 (0.0%)	4 (2.9%)		
Other	0 (0.0%)	4 (6.6%)	4 (2.9%)		
Primary Language				*χ*^2^ = 1.008	0.604
English	17 (22.7%)	10 (15.9%)	27 (19.6%)		
Spanish	57 (76.0%)	52 (82.5%)	109 (79.0%)		
Creole	1 (1.3%)	1 (1.6%)	2 (1.4%)		
Education Level				*χ*^2^ = 7.884	0.343
Elementary or less	1 (1.4%)	0 (0.0%)	1 (0.7%)		
Some High School	2 (2.7%)	2 (3.3%)	4 (3.0%)		
High School/GED	2 (2.7%)	2 (3.3%)	4 (3.0%)		
Technical Training	12 (16.2%)	14 (23.0%)	26 (19.3%)		
Some College	3 (4.1%)	9 (14.8%)	12 (8.9%)		
Associate Degree	18 (24.3%)	13 (21.3%)	31 (23.0%)		
Bachelor’s Degree	10 (13.5%)	6 (9.8%)	16 (11.9%)		
Graduate Degree	25 (33.8%)	14 (23.0%)	39 (28.9%)		

**Table 2 healthcare-12-02501-t002:** Multilevel model results predicting teachers’ observed post-intervention safety practice skills.

Level and Variable	β	SE	*p*-Value
Classroom level			
Baseline HERS-C Safety	0.329	0.127	0.010
Child disability ratio within classroom	−0.259	0.136	0.057
Intervention × child disability ratio	−0.098	0.104	0.345
Residual variance	0.830	0.094	<0.001
Childcare center level			
Intervention group	0.682	0.164	<0.001
Intercept	16.956	3.679	<0.001
Residual variance	0.526	0.217	0.016

Note: β = standardized coefficient; SE = standard error; intervention group (JS+CS = 1 and HC2 = 0). The dependent variable was HERS-Safety.

**Table 3 healthcare-12-02501-t003:** Multilevel model results predicting teachers’ observed post-intervention child behavior management skills.

Level and Variable	β	SE	*p*-Value
Classroom Level			
Baseline HERS-C Behavior	0.375	0.119	0.002
Child Disability Ratio within Classroom	−0.076	0.149	0.609
Intervention × Child Disability Ratio	−0.074	0.109	0.496
Residual Variance	0.828	0.089	<0.001
Childcare Center Level			
Intervention Group	0.561	0.157	<0.001
Intercept	11.060	1.867	<0.001
Residual Variance	0.680	0.173	<0.001

Note: β = standardized coefficient; SE = standard error; intervention group (JS+CS = 1 and HC2 = 0). The dependent variable was HERS-Safety.

**Table 4 healthcare-12-02501-t004:** Multilevel model results predicting teachers’ observed post-intervention classroom communication practices.

Level and Variable	β	SE	*p*-Value
Classroom Level			
Baseline HERS-C Communication	0.174	0.113	0.125
Child Disability Ratio within Classroom	−0.374	0.142	0.008
Intervention × Child Disability Ratio	0.030	0.109	0.785
Residual Variance	0.824	0.096	<0.001
Childcare Center Level			
Intervention Group	0.266	0.208	0.200
Intercept	8.810	1.516	<0.001
Residual Variance	0.918	0.114	<0.001

Note: β = standardized coefficient; SE = standard error; intervention group (JS+CS = 1 and HC2 = 0). The dependent variable was HERS-Communication.

**Table 5 healthcare-12-02501-t005:** Multilevel model results for observed post-intervention teacher resiliency coping.

Level and Variable	β	SE	*p*-Value
Classroom Level			
Baseline HERS-C Resiliency	0.257	0.116	0.027
Child Disability Ratio within Classroom	−0.265	0.168	0.115
Intervention × Child Disability Ratio	0.140	0.113	0.217
Residual Variance	0.848	0.095	<0.001
Childcare Center Level			
Intervention Group	0.439	0.169	0.010
Intercept	5.876	0.919	<0.001
Residual Variance	0.802	0.417	0.001

Note: β = standardized coefficient; SE = standard error; intervention group (JS+CS = 1 and HC2 = 0). The dependent variable was HERS-Resiliency.

**Table 6 healthcare-12-02501-t006:** Multilevel model results for post-intervention teacher self-efficacy.

Level and Variable	β	SE	*p*-Value
Classroom Level			
Baseline Teacher Self-Efficacy	0.033	0.110	0.766
Child Disability Ratio within Classroom	−0.008	0.153	0.958
Intervention × Child Disability Ratio	0.080	0.140	0.570
Residual Variance	0.973	0.035	<0.001
Childcare Center Level			
Intervention Group	0.306	0.239	0.201
Intercept	14.445	3.447	<0.001
Residual Variance	0.893	0.150	<0.001

Note: β = standardized coefficient; SE = standard error; intervention group (JS+CS = 1 and HC2 = 0). The dependent variable was post-intervention teacher self-efficacy.

**Table 7 healthcare-12-02501-t007:** Multilevel model results for post-intervention teacher-perceived job control.

Level and Variable	β	SE	*p*-Value
Classroom Level			
Baseline Teacher Job Control	0.229	0.092	0.013
Child Disability Ratio within Classroom	−0.059	0.130	0.649
Intervention × Child Disability Ratio	−0.017	0.103	0.867
Residual Variance	0.935	0.045	<0.001
Childcare Center Level			
Intervention Group	0.242	0.285	0.396
Intercept	9.900	3.189	0.002
Residual Variance	0.921	0.152	<0.001

Note: β = standardized coefficient; SE = standard error; intervention group (JS+CS = 1, and HC2 = 0). The dependent variable was post-intervention childcare worker job control.

## Data Availability

Requests for data can be sent to the corresponding author.

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
