# Peer review of "Using RE-AIM to Assess Infant Early Childhood Mental Health Practices in Classrooms Serving Children with and Without Disabilities"

_healthcare, 2024, doi:10.3390/healthcare12242501_

Round 1
Reviewer 1 Report
Comments and Suggestions for Authors
The article presents a topic of interest. There are issues that could be improved, especially in the presentation of the results, where the expression of the data obtained could be more ‘visual’, as well as the use of information flows and diagrams that could improve the understanding of the study. It is true that the theoretical framework is adequate. The writing is fluent, but I recommend that the number of references to inclusive education and attention to diversity in the last 3-4 years could be increased. In the case of the conclusions, I consider that this is the section that requires the most important improvements. First of all, it is not good to use any kind of references, but to investigate and propose more practical ideas on inclusive education. Secondly, it is necessary to allude to the relevance of ICT for the improvement of new educational formulas and initiatives for inclusiveness and personalisation of educational responses.
Author Response
Comment 1: There are issues that could be improved, especially in the presentation of the results, where the expression of the data obtained could be more ‘visual’, as well as the use of information flows and diagrams that could improve the understanding of the study.
Response 1: We thank the reviewer for this suggestion. We have added a conceptual figure to explain our hypothesized perception of how JS+CS impacts teacher outcomes and how this relationship may be moderated by classroom child disability ratio. With regard to the tables, we believe how the information presented provides full transparency of study findings and is presented in a manner that aligns with standard practice. Additionally, we recommend that the reviewer see our graphical abstract, which may provide an additional visual clarity. We are open to other specific suggestions that the reviewer may have about expressing the data in a more visual format.
Comment 2: I recommend that the number of references to inclusive education and attention to diversity in the last 3-4 years could be increased.
Response 2: We have included more information on inclusive education in early childhood programs in the first paragraph of the Introduction with two recent citations that highlight the importance of access and participation in these programs for children with disabilities.
Comment 3: In the case of the conclusions, I consider that this is the section that requires the most important improvements. First of all, it is not good to use any kind of references, but to investigate and propose more practical ideas on inclusive education. Secondly, it is necessary to allude to the relevance of ICT for the improvement of new educational formulas and initiatives for inclusiveness and personalization of educational responses.
Response 3: We appreciate this suggestion to include recommendations about information and communication technology and to make the conclusions more practical. It is our understanding that when information and communication technology is used to improve the functional capabilities of individuals with disabilities, it would be referred to as assistive technology. We therefore specify in the discussion and conclusions the potential for assistive technology to improve outcomes in inclusive classrooms.
Reviewer 2 Report
Comments and Suggestions for Authors
The article is well structured and has a solid theoretical foundation that supports the research component.
We believe that it would be useful if the conclusions were formulated somewhat more synthetically and more specifically so as to highlight even better the importance and impact of the present study.
Perhaps it would be worthy to be highlighted in the discussion chapter, but also formulated in the conclusions chapter some clear and specific ideas from which to derive the concrete impact on the child after the use of the Model in the assessment of the mental health of the early infant.
Author Response
Comment 1: We believe that it would be useful if the conclusions were formulated somewhat more synthetically and more specifically so as to highlight even better the importance and impact of the present study.
Response 1: Thank you for the suggestion to improve the clarity of the conclusions. We re-wrote this section to better highlight the impact of our findings.
Comment 2: Perhaps it would be worthy to be highlighted in the discussion chapter, but also formulated in the conclusions chapter some clear and specific ideas from which to derive the concrete impact on the child after the use of the  Model in the assessment of the mental health of the early infant.
Response 2: We note in the limitations that this study examines immediate post-intervention effects, and longer-term effects are unknown. We appreciate the reviewer's suggestion to specifically address the impact on children, which will be examined in the future as we collect additional data. We now note this in the limitations and future directions.
Reviewer 3 Report
Comments and Suggestions for Authors
Great paper - I think it could be improved by clarifying or justifying analytical decisions or adjusting some of the analysis.
Methods
Please correct the Procedures description of data collection every 6 months for 24 months, given that the intervention was shortened to 14 weeks.
Cite the ecological validity model used to develop the toolkit.
Analysis and Results
It is not initially clear in your analysis that you are using multi-level modeling because you say "multiple regression."
Throughout the paper you use "between and within" in ways that are typical of ANOVA models, but not multilevel models. Consider using "Level 1" for your smallest unit of analysis (classroom/teacher)and "Level 2" for the cluster. Another option could be "Classroom level" and "Center level". Perhaps you mean "within" as in "within the center", but using "between" for the center level was confusing as a reader.
You mention the teacher outcome variable was included at both the between and within levels. I'm not sure what that would mean and I don't see any clarification of the statement in your tables.
It looks like you are running the same model for each teacher outcome - please include a representation of the formula at least once.
Please clarify in your table labels where a score is baseline or "pre-" score predicting a post-score.
Are you interested only in the moderation of classroom composition on outcomes? Or is the influence of classroom disability ratio interesting by itself? If it is of interest, consider showing results without the interaction term, as retaining it may mask the influence of disability ratio.
Conclusions
The REAIM model contains results that might be more interesting in analysis, which it looks like you began to do - as 4.3 and 3.3. seem redundant. In that section, you mention a 30% attrition - this should also be mentioned in your Limitations.
I find this claim problematic - "The program's impact on teachers effective communication practices with higher proportions of children with disabilities requires further investigation." This indicates that there was a significant moderation because it discusses impact in the context of differing proportions. However, while the influence of disability ratio was evident, this was not moderated by the intervention - as indicated by the non-significant interaction term.
Author Response
Methods
Comment 1: Please correct the Procedures description of data collection every 6 months for 24 months, given that the intervention was shortened to 14 weeks. 
Response 1: Thank you for catching this error. We have corrected this error in the Procedures section.
Comment 2: Cite the ecological validity model used to develop the toolkit.
Response 2: We added a reference to O’Connor et al. (2020).
Analysis & Results
Comment 3: It is not initially clear in your analysis that you are using multi-level modeling because you say "multiple regression."
Response 3: Thank you for catching this error. This has been corrected at the beginning of the data analytic section.
Comment 4: Throughout the paper you use "between and within" in ways that are typical of ANOVA models, but not multilevel models.  Consider using "Level 1" for your smallest unit of analysis (classroom/teacher) and "Level 2" for the cluster. Another option could be "Classroom level" and "Center level". Perhaps you mean "within" as in "within the center", but using "between" for the center level was confusing as a reader.
Response 4: Thank you for articulating this. We have revised the analytic section, the results, and each of the tables so that the levels are explained at the classroom level and the childcare center level. We believe this suggestion makes it much easier for the reader to interpret the findings. Thank you again for this helpful suggestion.
Comment 5: You mention the teacher outcome variable was included at both the between and within levels. I'm not sure what that would mean and I don't see any clarification of the statement in your tables. 
Response 5: Thanks for pointing out this confusing statement. We have corrected this statement to say the following: The teacher outcome variable was included at the classroom level.
Comment 6: It looks like you are running the same model for each teacher outcome - please include a representation of the formula at least once. 
Response 6: We have attempted to provide additional explanation for the models that were conducted within Mplus using the following: To account for the nested structure of teachers within programs within each analysis, we employed a two-level random effects model using maximum likelihood estimation with Monte Carlo integration (500 iterations). At Level 1 (within-program level), we included teachers' baseline classroom variables (group-mean centered), child disability ratio, and the interaction between intervention condition and child disability ratio (group-mean centered). At Level 2 (between-program level), we included intervention condition (grand-mean centered) as a program-level predictor.
Comment 7: Please clarify in your table labels where a score is baseline or "pre-" score predicting a post-score.
Response 7: Thanks for catching this. Within the table titles, we have added language about it being a post score and within the table, we have added “Baseline” before each teacher predictor variable.
Comment 8: Are you interested only in the moderation of classroom composition on outcomes? Or is the influence of classroom disability ratio interesting by itself? If it is of interest, consider showing results without the interaction term, as retaining it may mask the influence of disability ratio. 
Response 8: Within this study, our hope was to find that JS+CS worked equitably regardless of child disability classroom ratio so we were specifically interested in the moderation of classroom composition on outcomes.
Conclusions
Comment 9: The REAIM model contains results that might be more interesting in analysis, which it looks like you began to do - as 4.3 and 3.3. seem redundant. In that section, you mention a 30% attrition - this should also be mentioned in your Limitations. 
Response 9: Within 4.3, we attempted to just briefly mention the limited dosage finding and provide some possible explanations for why implementation was not conducted with the intended fidelity. We are in agreement with the reviewer that mention and explanation of the high attrition rate should be mentioned in the Limitations. We have added the following: The high immediate teacher attrition rate in both the JS+CS and the HC2 groups (no consultations received) limited our understanding of the effects of the intervention. Further, systematic exploration of why teachers were lost to follow-up was not conducted. However, anecdotally some childcare centers completely dropped from the study due to concerns about measure burden and time commitment of participating in ongoing consultation, when they were already short-staffed.
Comment 10: I find this claim problematic - "The program's impact on teachers effective communication practices with higher proportions of children with disabilities requires further investigation." This indicates that there was a significant moderation because it discusses impact in the context of differing proportions. However, while the influence of disability ratio was evident, this was not moderated by the intervention - as indicated by the non-significant interaction term. 
Response 10: This statement has been removed from the conclusions section.